# Aerobiological Monitoring and Metabarcoding of Grass Pollen

**DOI:** 10.3390/plants12122351

**Published:** 2023-06-17

**Authors:** Anastasia A. Krinitsina, Denis O. Omelchenko, Artem S. Kasianov, Vera S. Karaseva, Yulia M. Selezneva, Olga V. Chesnokova, Vitaly A. Shirobokov, Svetlana V. Polevova, Elena E. Severova

**Affiliations:** 1Department of Higher Plants, Faculty of Biology, Lomonosov Moscow State University, 119991 Moscow, Russia; krinitsina@msu-botany.ru (A.A.K.); chesnokovaov@my.msu.ru (O.V.C.); vitalijsirobokov41@gmail.com (V.A.S.); polevova@mail.bio.msu.ru (S.V.P.); 2Laboratory of Plant Genomics, Institute for Information Transmission Problems, 127051 Moscow, Russia; omelchenkodo@iitp.ru (D.O.O.); artem.kasianov@gmail.com (A.S.K.); 3Department of Biology, Institute of Natural Science, S.A. Esenin Ryazan State University, 390000 Ryazan, Russia; v.karaseva94@mail.ru (V.S.K.); posevina_julia@mail.ru (Y.M.S.)

**Keywords:** ITS1, ITS2, airborne pollen, metabarcoding, Poaceae, phenology, Russia

## Abstract

Grass pollen is one of the leading causes of pollinosis, affecting 10–30% of the world’s population. The allergenicity of pollen from different Poaceae species is not the same and is estimated from moderate to high. Aerobiological monitoring is a standard method that allows one to track and predict the dynamics of allergen concentration in the air. Poaceae is a stenopalynous family, and thus grass pollen can usually be identified only at the family level with optical microscopy. Molecular methods, in particular the DNA barcoding technique, can be used to conduct a more accurate analysis of aerobiological samples containing the DNA of various plant species. This study aimed to test the possibility of using the ITS1 and ITS2 nuclear loci for determining the presence of grass pollen from air samples via metabarcoding and to compare the analysis results with the results of phenological observations. Based on the high-throughput sequencing data, we analyzed the changes in the composition of aerobiological samples taken in the Moscow and Ryazan regions for three years during the period of active flowering of grasses. Ten genera of the Poaceae family were detected in airborne pollen samples. The representation for most of them for ITS1 and ITS2 barcodes was similar. At the same time, in some samples, the presence of specific genera was characterized by only one sequence: either ITS1 or ITS2. Based on the analysis of the abundance of both barcode reads in the samples, the following order could describe the change with time in the dominant species in the air: *Poa*, *Alopecurus*, and *Arrhenatherum* in early mid-June, *Lolium*, *Bromus*, *Dactylis*, and *Briza* in mid-late June, *Phleum*, *Elymus* in late June to early July, and *Calamagrostis* in early mid-July. In most samples, the number of taxa found via metabarcoding analysis was higher compared to that in the phenological observations. The semi-quantitative analysis of high-throughput sequencing data well reflects the abundance of only major grass species at the flowering stage.

## 1. Introduction

The grass family (Poaceae) is the second largest monocotyledonous family with more than 11,000 species [1,2], of which about 270 species are found in central Russia [3]. Most grasses are wind-pollinated [4,5] and have high pollen production [6,7,8,9,10,11]. Grass pollen is the leading aeroallergen worldwide; about 10–30% of the world’s population is affected by it [12,13,14]. The allergenicity of pollen of grass species is not the same, ranging from moderate to very high [15,16], and flowering periods often overlap. The pollen of *Phleum* spp., *Dactylis* spp., *Lolium* spp., *Trisetum* spp., *Festuca* spp., *Poa* spp., *Cynodon* spp., and *Anthoxanthum* spp. is the most common cause of hay fever in the Northern Hemisphere [13]. Despite considerable cross-reactivity [17], allergens differ between grass species [13,18]. Thirteen groups of allergens have been identified in grass pollen [13], of which allergens of groups 1 and 5 are responsible for developing an allergic reaction in 95% of patients suffering from hay fever [19]. Currently, the most effective way to prevent the development of allergies is considered to be allergen-specific immunotherapy (AIT), known as “allergy vaccinations”, in which it is extremely important to control both the dosage of the allergen and its specificity. Accurate identification of the source of allergenic proteins is critical due to the specificity of the immune response they elicit in patients depending on their sensitization profile [20,21,22,23,24,25,26].

Aerobiological monitoring is a necessary element of a complex of activities that allows one to track and predict the dynamics of the concentration of the main allergens in the air and adjust the therapy for and lifestyle of patients with hay fever [12]. Light microscopy is the standard method for identifying pollen in air samples [27]. However, the pollen of all species of the Poaceae family belongs to the same palynomorphological type and is not distinguishable even down to the genus level during routine aerobiological analysis [28]. Therefore, standard pollen calendars usually represent grasses as a single group.

The detailing of pollen curves is possible using molecular methods or based on phenological observations. Phenological analysis has already been used to interpret aerobiological data for trees, shrubs [29,30,31,32] and herbaceous plants [33,34,35,36,37,38]. The combination of phenological data with aerobiological analysis can help associate symptoms with the pollination of specific taxa. However, phenological observations in aerobiology are scarce, as they require deep botanical knowledge. In addition, phenological observations, as a rule, do not allow a quantitative assessment of the participation of individual species in the formation of the pollen spectrum. One way to solve this problem was proposed by Ghitarrini et al. [38]. They proposed for the calculation of each species’ phenological index, which is the product of the point value of the phenophase, the abundance of the species, and its pollen production. A comparison of the concentration curve and the dynamics of the phenological index carried out in Perugia during one season showed a high correlation between these two parameters.

Problems of species identification in aerobiological studies can be solved using molecular diagnostic methods. In the last decade, several works have appeared that show the promise of using DNA analysis methods on air samples to determine the dynamics of the composition of the pollen spectrum [39,40,41,42,43].

Longhi et al. [44] showed the possibility of replacing routine analysis with light microscopy methods with quantitative PCR analysis but did not test the method on real monitoring data. Successful detection of three plant species individually in pollen samples with TaqMan probes and a good correlation between PCR cycle and pollen grain abundance by Longhi et al. [43] showed the possibility of replacing routine analysis with light microscopy methods with quantitative PCR analysis. Other studies showed that the two methods are complementary. There are some taxa that could not be detected using one method but could be detected by the other [45]. It seems that much work is still needed to be carried out to develop a reliable standard protocol. Mohanty, Buchheim and Levetin [46], Leontidou et al. [41], and Ghitarrini et al. [47] proposed a method for assessing the composition of the airborne pollen spectrum using species-specific PCR for known plastid genes. This approach shows promising results in the analysis of pollen dynamics; however, it is limited by a number of species predefined for PCR analysis and does not allow the identification of the full spectrum of species present in a pollen sample.

The plastid regions rbcL and trnL, as well as the ITS1 and ITS2 regions of nuclear ribosomal internal transcribed spacers (nrITS), are most used for pollen metabarcoding. For complex aerobiological samples containing pollen from various plants, fungal spores, bacteria, and viruses, the correlation between pollen concentration and DNA reads using the rbcL plastid marker turned out to be relatively low [48,49]. Kraaijeveld et al. [40] proposed a high-throughput trnL sequencing method, which in some cases made it possible to succeed in the generic identification of grass pollen, but this method has not yet been tested on a large dataset. Korpelainen and Pietilainen [50] used ITS2 sequence metabarcoding to study the composition and dynamics of the indoor pollen spectrum and its potential impact on human health. The works of Johnson et al. [51], Polling et al. [52], Leontidou et al. [41], and Campbell et al. [48] showed the possibilities of the DNA metabarcoding method for a qualitative assessment of the composition of the pollen spectrum from air samples. All researchers noted that the taxonomic diversity identified using molecular methods is significantly higher than that identified in routine aerobiological analysis. Based on the analysis of aerobiological samples from the Netherlands, Polling et al. [52] suggested that nrITS2 should be the preferred marker of choice for the molecular monitoring of pollen in the air.

Most works on the metabarcoding of air samples set themselves the task of identifying the pollen of plants from different families. However, attempts to use this method to identify pollen from plants of the same family are extremely rare. Brennan et al. [42] used cyclone-type pollen traps and rbcL and ITS2 marker sequences to estimate the spatial–temporal distribution of grass pollen during one growing season in the UK. They showed that the method works well at the generic level, allowing not only a determination of the qualitative composition of the spectrum but also the provision of a semi-quantitative estimate of pollen occurrence. Campbell et al. [48] used the rbcL marker sequence and identified the main grass species responsible for pollen concentration peaks throughout the pollination season.

In addition to the nuclear regions of ITS1 and ITS2, a promising nuclear marker for the analysis of pollen samples is the ETS marker sequence, which has the highest specific information content among nuclear and plastid barcodes in the Poaceae family [53,54,55]. A comparison of the efficiency of ITS1, ITS2, ETS, and trnL barcodes, carried out on artificial mixtures of grass pollen [55], showed that nuclear barcodes are more effective than plastome ones are, both in the amplification of pollen DNA and in the identification of grass species. The results of metabarcoding corresponded to the composition of pollen mixtures at a qualitative level. However, the results of the quantitative assessment based on the count of reads did not correspond to the actual ratio of pollen grains of different species in mixtures.

This study aimed to test the possibility of using the ITS1 and ITS2 nuclear loci for determining grass pollen from air samples via metabarcoding and to compare the analysis results with phenological observations.

## 2. Results

### 2.1. Sequencing Results

Based on the high-throughput sequencing data of 131 samples, the average number of reads per sample for ITS1 and ITS2 regions were 95,997 and 66,179, respectively, of which 5061 and 1398 reads on average (5.3% and 2.1%) were mapped to our local database of reference barcode sequences of the grass species common in the Moscow and Ryazan regions.

In four samples, the number of total ITS2 reads was small (from 10 to 160). Three of them demonstrated very poor amplification efficiency; after purification of the amplification product, the DNA concentration was below the sensitivity limit (less than 0.1 ng/µL). For one sample, a small number of ITS1 reads were obtained (108 in total).

Based on the high-throughput sequencing data for two regions of the nuclear genome ITS1 and ITS2, ten genera of the Poaceae family were detected in the 131 studied airborne pollen samples.

For the genera *Dactylis, Poa, Lolium*, *Calamagrostis, Alopecurus, Arrhenatherum, Phleum,* and *Briza*, either both marker sequences or only ITS1 reads were predominantly detected, but in some cases (1–2 samples for each genus) only ITS2 sequences were identified. The proportion of samples in which a particular genus was detected only through ITS1 was rather high and could reach 35% (*Arrhenatherum*) (Figure 1).

For two genera, *Elymus* and *Bromus*, the number of samples in which pollen was detected only through ITS1 turned out to be less than the number of samples in which only ITS2 sequences were found (Figure 1). Of the 61 samples where *Bromus* reads were detected, 5 samples contained only ITS1, 11 samples contained only ITS2 and 74% (45 samples) had reads of both sequences. For *Elymus*, more than half (54%) of the samples contained only ITS2 reads, and only one contained ITS1 sequences. In those samples where reads of both regions were found, the representation of ITS1 was significantly lower (up to 20 times) than that of ITS2.

### 2.2. Metabarcoding Analysis of Airborne Samples from Moscow and Ryazan

Based on the analysis of the abundance of both barcode reads in the samples, the change in the dominant species of the aerobiological spectrum during the pollen season was the following (Figure 2 and Appendix A). In the first half of June, barcodes of *Poa* were detected in the samples of all years in both locations. The abundance of ITS1 and ITS2 reads of the genus *Poa* decreased from 88–91% at the beginning of the pollen season to 8% by its end. The opposite tendency was observed for *Calamagrostis*; the abundance of both barcodes was minimal at the beginning of the pollen season and reached up to 50–67% (Moscow) and 99% (Ryazan) by its end. The abundance of ITS1 and ITS2 barcodes of *Dactylis* and *Lolium* gradually increased during the first ten days and reached its maximum during the next ten days of the observation period (from the end of June to the first ten days of July) and then gradually decreased by the end of July. *Bromus*, *Elymus,* and *Phleum* reads were detected since the 7–10th day of the observation period, but with less abundance. The total amount of reads of *Elymus* was very low in all samples; the abundance of *Bromus* reads was higher in Ryazan (46%) compared to that in Moscow (6%). *Arrhenatherum* barcodes were detected during the first ten days of observation in Moscow in 2020 and 2021 and in Ryazan in all seasons but in very small amounts (less than 5%). *Alopecurus* reads of both markers were also detected at the beginning of the observation period in all seasons but its abundance did not exceed 3–4%. Barcodes of *Briza* were identified in very small amounts in Moscow in 2020 and 2021 and in Ryazan in all seasons.

Metabarcoding analysis of aerobiological samples collected in Moscow and Ryazan in 2020–2022 showed an identical taxonomic composition of pollen spectra in the two locations and a similar order of changes. The overall similarity of the sequence of flowering taxa is high (Spearman rank correlation coefficient = 0.86, *p*-value = 0.002), and the similarity in some parts of the series is slightly lower (Kendall tau rank correlation coefficient = 0.75, *p*-value = 0.003). The spectra of the two locations differed quantitatively; the abundance of reads of *Arrhenatherum* was 10 times higher in Moscow (18%) compared to that in Ryazan. On the contrary, the number of reads of *Bromus*, *Elymus*, *Calamagrostis,* and *Phleum* was higher in the samples from Ryazan. The largest number of reads at both locations belonged to three taxa *Poa*, *Calamagrostis,* and *Lolium*. In Moscow, their abundance ranged from 62% to 100%, and in Ryazan, it tanged from 45% to 99,7%. The taxonomic composition of samples was the most diverse in the middle of the pollen season, on days 12–20 of observations.

### 2.3. Comparison of Metabarcoding and Phenological Analysis

The metabarcoding analysis and phenological data were compared on 28 samples (Figure 3 and Figure 4). In most cases (86%—24 samples out of 28), the range of taxa detected via metabarcoding was wider than the range of flowering species observed in the sample plots. The DNA of all species that bloomed in the sample plots was identified via metabarcoding. The exceptions were *Briza* (in Moscow and Ryazan, not observed in the sample plots), *Alopecurus* (in Moscow), and *Arrhenatherum* (in Ryazan). The abundance of reads was not consistent with the values of the phenological index.

## 3. Discussion

In many, studies the nuclear barcode ITS2 is suggested for the taxonomic identification of plants in aerobiological samples [48,50,51,52]. However, comparing the read abundance of both barcode sequences (ITS1 and ITS2) showed that ITS2 is not the best choice for some of the grass genera. *Alopecurus, Phleum,* and especially *Bromus* and *Elymus* were stably identified using the ITS2 barcode in the airborne pollen samples. On the other hand, the use of ITS2 only could lead to false negative results for *Lolium*, *Dactylis*, and *Calamagrostis* detection, for which no ITS2 sequences were found in a large number of samples, while the abundance of ITS1 reads of these genera was high. Moreover, generally, the proportion of ITS1 and ITS2 reads for *Briza*, *Bromus*, *Elymus*, *Arrhenatherum*, and *Alopecurus* was much lower than that for species of *Poa* and *Calamagrostis*. A similar tendency to underestimate or overestimate the abundances of some species in pollen mixtures in combination with rbcL and ITS2 barcodes was also observed by Bell et al. [56]. We agree with Bell et al. that there could be several sources of these biases: DNA barcode copy number, efficiency of DNA isolation from pollen of different species, and PCR efficiency due to structural features of the analyzed genome regions. We suggest that interspecies polymorphism and nucleotide composition affect the PCR efficiency of these barcodes in complex pollen mixtures. For example, the presence of GC-rich regions in the *Elymus* ITS1 sequence [55] could lead to the lower efficiency of PCR of this barcode in pollen mixtures and, thus, its underrepresentation in the results.

Based on the analysis of both barcodes in the airborne pollen samples, the change in the dominant species in both locations can be described by the following sequence: *Poa, Alopecurus,* and *Arrhenatherum* in early mid-June, then *Lolium*, *Bromus*, *Dactylis* in mid-late June, then *Phleum*, *Elymus,* and *Briza* at the end of June to the beginning of July, and finally *Calamagrostis* in early mid-July. The similarity of the sequences of flowering taxa at both locations is large and significant. Despite the differences in the composition of the surrounding vegetation, we failed to identify differences in the taxonomic composition of the spectra. All differences were quantitative; the abundance of *Arrhenatherum* barcodes was higher in the samples from Moscow, and, on the contrary, reads of *Bromus*, *Elymus*, *Calamagrostis,* and *Phleum* were more abundant in the samples from Ryazan. Moscow samples are dominated by grasses widely used for lawns with different species of *Poa, Lolium* and *Arrhenatherum*. The samples from Ryazan are dominated by meadow grasses, which are widespread in the Oka floodplain. Artificial lawns are not common in Ryazan.

When comparing phenological, aerobiological, and metabarcoding data, it should be taken into account that the abundance of reads in a sample reflects neither the pollen concentration in the air nor the phenological index. A small number of reads (in our study for *Bromus* and *Elymus*) may be associated with amplification problems and reflect, for example, the features of the DNA nucleotide composition rather than the abundance of pollen of these taxa in the air sample. Thus, phenological and metabarcoding data should be compared at the descriptive level.

In our study, the phenological and metabarcoding analysis showed a similar order of grass flowering during summer. In most cases, the number of taxa found via metabarcoding analysis was higher than that found via phenological observations (Figure 3 and Figure 4). The discrepancies were most often associated with the identification of *Bromus inermis* and *Phleum pratense*. The results obtained in this work are congruent with the conclusions of Campbell et al. [48] and Brennan et al. [42] about the possibility of using high-throughput sequencing data for semi-quantitative analysis. Such semi-quantitative analysis well reflects the representation of the major species. The high-throughput sequencing data could deliver only qualitative results for species with a low pollen representation in a sample (species that were not at the peak of flowering) or if the barcode region had features that reduce amplification efficiency.

Higher taxonomic diversity in aerobiological samples detected using molecular methods compared to aerobiological analysis has been noted in many studies primarily due to the peculiarities of the formation of the air pollen spectrum. A pollen trap installed 10–12 m above ground level on the Central Russian Plain could reflect pollen diversity within a radius of 50 km [57]. However, phenological observations made at one point did not reflect flowering over such a large area [32]. This was clearly seen in the analysis of specific samples in our study. During the first year of work (2020) in Ryazan, phenological observations were carried out at three sites near the pollen trap. *Calamagrostis epigejos* was absent in those sample plots; however, its DNA was found in the airborne pollen samples. An increase in the number of sample plots in 2021 and 2022 showed that the discovery of *Calamagrostis* DNA coincided with the flowering of this species in the remote sample plots. The detection of *Briza* DNA in air samples was also associated with the flowering of this species outside the sample plots; in Moscow, *Briza media* grows on the territory of the Botanical Garden of Moscow State University, and in Ryazan, *Briza media* is a common species in floodplain meadows around the city [3]. The time of *Briza* DNA detection in airborne pollen samples corresponded to the flowering time of *Briza media* [3].

The pollen spectrum may contain pollen, which secondarily lifts in the air after the completion of flowering. The secondary rise and transport of pollen from distant regions can explain the detection of DNA of some taxa after the completion of their flowering in the phenological sample plots. For example, it could explain the detection of *Alopecurus* and *Arrhenatherum* DNA in pollen samples at the end of June–July or the wide range of barcodes in air samples at the end of pollen season, when phenological observation showed only *Calamagrostis epigejos* (Figure 3).

Thus, it is advisable to use both barcodes (ITS1 and ITS2) to analyze pollen in air samples. This makes it possible to carry out a semi-quantitative analysis, trace the dynamics of changes in the qualitative composition of the pollen spectrum and determine the dominant species. However, a quantitative analysis of the mixture is impossible since it is necessary to produce DNA fragments for further sequencing using PCR during sample preparation. Furthermore, due to the peculiarities of the structure of the genomes in different grass species, in particular, interspecific polymorphism and nucleotide composition, the efficiency of amplification for them is different. As a result, PCR is less efficient for “complex” matrices, the proportion of amplicons of some species is artificially overestimated in the resulting mixture and the representation of others is reduced. Using PCR-free library preparation protocols for airborne pollen samples is impossible due to the low total concentration of grass pollen in the air, leading to an insufficient DNA yield. The problem could be solved by changing the sampling method, i.e., using high-volume impactors or combining several samples into one. For a more precise systematic determination, it is necessary to look for additional barcodes that increase identification accuracy.

## 4. Materials and Methods

### 4.1. Sampling

The study was carried out at two points: in Moscow (55°42′ N, 37°32′ E) and Ryazan (54°37′ N, 39°41′ E) in 2020–2022 during June–July using standard Hirst-type volumetric pollen traps [58]. Both cities are located within the same bio-geographical zone; the distance between them is 184 km. The entire area around Moscow is heavily urbanized, densely built up, and populated. Ryazan is located in the floodplain of the Oka River and is surrounded by floodplain meadows, which differ from the Moscow region in terms of the grass species composition and the abundance of individual species [3]. Pollen traps were installed in open areas on the roofs of buildings 12 m high (Moscow) and 20 m high (Ryazan). Two pollen traps were installed at each observation point. The pollen samples from one trap were analyzed using light microscopy, and pollen concentration of grasses was determined daily according to the standard method [27]. The samples from the second trap were used for DNA extraction. The metagenomic analysis included only samples in which the daily concentration of grass pollen was at least 50 pollen grains/m^3^ according to aerobiological observations. In total, 131 samples (days) were obtained over three years of observations (Moscow, 47 samples; Ryazan, 84 samples. Appendix A). The different number of samples at two points reflects the different intensity of grass flowering in Moscow and Ryazan.

#### Phenological Observations

Phenological observations were carried out according to the method proposed by Ghitarrini et al. [38]. Sample plots at least 100 m^2^ in size were placed around each point of the trap location (five plots in Ryazan and nine plots in Moscow). During the first year of the study, three plots were selected in Ryazan and seven plots were chosen in Moscow in the close vicinity (within 1 km radius) of the pollen trap locations. The difference in the number of sample plots was due to the presence of unmowed areas suitable for observation. During the second and third years, two remote (within 25 km) sample plots were added at each point. A list of species growing in each plot was compiled (Appendix A) and their abundance was calculated. The abundance of the species was estimated in a simplified way (dominant/non-dominant) after Ghittarini et al. [38]. We considered species with a total cover surface of 25% or more to be dominant. The species’ phenological state was assessed based on the BBCH scale [59] once every 4–7 days from 1 June to early August until the end of flowering of all grasses in the sample plots. The phenological phase was determined via observations of at least 25 individuals of each species. In the case of rare species, they were determined by all their representatives on the site. The phenological index of each species on each day of observation and the total phenological index was calculated according to the method proposed by Ghitarrini et al. [38] and modified by us. As all grass species on the sample plots were perennial, we used the actual pollen production of species instead of scoring it. Pollen production was previously determined [11] separately for each observation point.

### 4.2. Statistical Methods

Metagenomic data for three years were averaged over the entire observation period for each location. The beginning of observations was considered the day when the grass pollen concentration exceeding 50 pollen grains/m^3^ was recorded for the first time. To compare changes in the taxonomic composition of the spectra at two locations, nonparametric Spearman and Kendall rank coefficients were used. In each sample, the species were ranked in order of flowering peak, which was defined as the period with the highest abundance of reads. Data analyses were performed in R 4.0.5 [60].

### 4.3. Optimization of Pollen Wash-off Procedure

Pollen was washed off the tapes into Longmire’s buffer with SDS (0.1 Tris-HCl, pH 8.0; 0.1 M EDTA, pH 8.0; 0.01 M NaCl; 0.005% SDS). Samples were incubated in the buffer on a rotator with a vertical rotation of 180° at a rotation speed of 20 rpm with short-term vibration in the extreme position. Mixing in the buffer ranged from 5 to 30 min in increments of 5 min. We used samples with artificially applied pollen to assess the optimal incubation time. First, dry pollen of *Phleum pratense* was applied on a sticky surface of Melinex tape. Then, the tape was divided into two parts; the first was used as a control and the second was placed into the buffer. The amount of pollen was calculated before and after washing according to the method applied for aerobiological slides.

For further extraction procedures, 15 min was selected as the optimal mixing time. After centrifugation, the pellet was washed once in 1 mL of Longmire’s buffer without SDS, centrifuged at room temperature at 2400 rpm for 1 min, and stored in a small amount of liquid at −70 °C.

### 4.4. Optimization of Pollen DNA Isolation

Four different methods were tested in the search for the best method for isolating DNA from pollen. Six samples were used to test the first, second, and third isolation methods, and eighteen samples were used to test the fourth method (six for each combination of enzymes). Dry pollen of *Phleum pratense* was used for testing.

DNA isolation from pollen according to the protocol described in [55]. The approximate time to isolate DNA from six samples is 4 h.Enzymatic treatment with lysozyme and zymolysin. First, the enzymatic treatment of pollen was carried out with the addition of 50 μL of lysozyme and 50 μL of zymolysin (1 U each) at a temperature of 37 °C for an hour. Then, 5 µL of proteinase K was added to the solution and incubated at 60 °C for 20 min without the homogenization stage. Next, 800–900 µL of lysis buffer (CTAB, 0.04% SDS) was added up to 1 mL of the final solution and incubated for another hour at a temperature of 60 °C. Then, DNA was extracted by adding a 1× volume of chloroform, stirring, centrifugation at room temperature for 30 min at 13,400 rpm, and collecting the upper aqueous fraction, which was placed in a clean tube. Next, DNA was precipitated by adding an equal volume of isopropanol and 0.1× *v*/*v* potassium acetate, followed by centrifugation at 4° C for 30 min at 15,000 rpm. Then, the DNA pellet was washed twice with 700 µL of 70% ethanol and dissolved in 20 µL of nuclease-free water. The approximate time to isolate DNA from six samples is 5 h and 15 min.Enzymatic treatment with lysozyme and zymolysin followed theisolation protocol from [55], excluding mechanical treatment (which combines the first and second methods). The approximate time to isolate DNA from six samples is 4 h and 45 min.

Enzymatic treatment with homogenization and standard phenol–chloroform isolation method. Pollen treatment was carried out by adding 1 U of each of the enzymes to the solution in three different combinations: 4a) chitinase; 4b) lysozyme + zymolysin; 4c) lysozyme + zymolysin + chitinase. Then, the samples were incubated at 37 °C for one hour. Further, pollen was homogenized using the Precellys Bacteria lysing kit CK01 (Bertin Technologies, Montigny-le-Bretonneux, France) on a Minilys instrument (Bertin Technologies, Montigny-le-Bretonneux, France) at a maximum speed of 5000 rpm for 240 s twice. Then, 10 µL of proteinase K was added to the solution and incubated at 60 °C for 20 min. Next, 600 µL of lysis buffer (CTAB, 0.04% SDS) was added to the solution and incubated for another hour at 60 °C. Finally, DNA extraction and purification were performed using the standard phenol–chloroform method. The approximate time to isolate DNA from six samples is 4 h and 45 min.

The purity of the DNA samples was assessed according to the A260/280, and A260/230 ratios on a NanoPhotometer N60-Touch (Implen, Munich, Germany), and the concentration was measured through fluorescence intensity using Qubit dsDNA HS Assay Kit (Invitrogen, Waltham, MA, USA) and a Qubit 3.0 fluorometer (Invitrogen, Waltham, MA, USA).

The fourth (4c) method was selected as optimal and used for all pollen DNA extractions in this work. This DNA extraction method made it possible to obtain the purest DNA preparations with the largest yield of DNA (Table 1). A260/280 and A260/230 ratios were within the range of pure DNA values of −1.8–2.0 and 2.0–2.2 (1.89 ± 0.13 and 2.09 ± 0.75, respectively), unlike those of DNA samples obtained using other methods. The A260/230 ratio of DNA samples obtained using methods 1–4b did not exceed 0.78 ± 0.52, which may indicate the residual presence of salts, phenols, proteins, or lipids. The A260/280 ratio was approximately 1.8 in almost all DNA samples, except for samples obtained using method 4a. The average DNA concentration in the samples obtained using the 4c method was also the highest, amounting to 1.13 ± 0.23 ng/µL. Additionally, PCR was performed with all obtained DNA samples using primers and a protocol that was used further for the amplification of marker regions. PCR products of the correct size were obtained only with DNA obtained using the 4c method. For the rest of the samples, obtaining stable amplification results suitable for library preparation was impossible.

### 4.5. PCR and Sequencing

NGS libraries were prepared using two-stage PCR [61,62] using ITS1 and ITS2 primers fused with Illumina Nextera adapters to simplify library preparation, which indicated the following: the primer for ITS1 (forward 5′-TCGTCGGCAGCGTCAGATGTGTATAAGAGACAGGGAAGGAGAAGTCGTAACAAGG-3′; reverse 5′-GTCTCGTGGGCTCGGAGATGTGTATAAGAGACAGAGATATCCGTTGCCGAGAGT-3′) and the primer for ITS2 (forward 5′-TCGTCGGCAGCGTCAGATGTGTATAAGAGACAGATCGAGTYTTTGAACGCAAGTTG-3′; reverse 5′-GTCTCGTGGGCTCGGAGATGTGTATAAGAGACAGTCCTCCGCTTATTGATATGCT-3′). The first PCR of pollen DNA samples conducted to obtain ITS1 and ITS2 amplicons was performed using Encyclo Plus PCR Kit (Evrogen, Moscow, Russia). The second PCR for library indexing was performed with the NEBNext Ultra II Q5 Master Mix (NEB, Ipswich, MA, USA) kit and Nextera XT Index Kit v2 Set A (Illumina, San Diego, CA, USA). After each amplification, DNA was purified using AMPure beads with a 1.1× bead ratio. Purified amplicons were quantified via fluorimetry using Qubit dsDNA HS Assay Kit and a Qubit 3.0 instrument.

High-throughput sequencing was performed on the Illumina MiSeq platform with MiSeq Reagent Kit v3, 2 × 300 nt paired-end (Illumina, San Diego, CA, USA).

### 4.6. Bioinformatics Analysis

Taxonomic analysis was preceded by quality trimming and the removal of adapters from raw sequencing reads using fastp v.0.23.2 software [63] with the parameters “cut_mean_quality 25 cut_right_window_size 10 cut_right_mean_quality 25”. ITS primer sequences were also trimmed using cutadapt v.4.2 software [64]. Next, taxonomic classification was carried out using the BLAST-based bioinformatic pipeline using the local grass barcode reference database described elsewhere [55,62] with a sequence similarity threshold of ≥99%, and an e-value of ≤0.01. Taxa with an abundance of less than 1% for all barcodes in each sample were discarded from the analysis.

## Figures and Tables

**Figure 1 plants-12-02351-f001:**
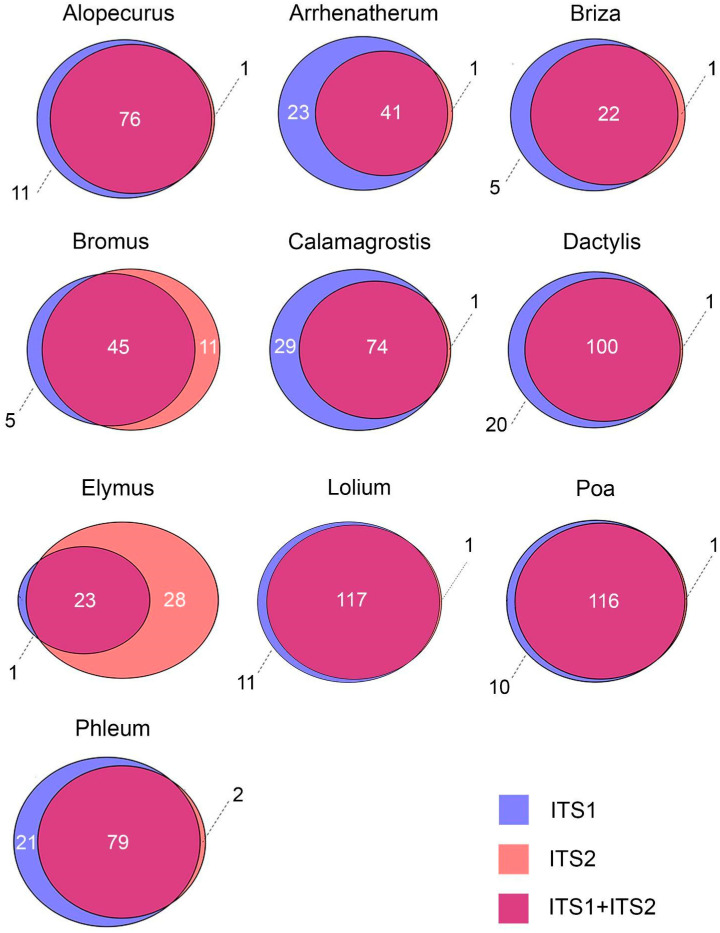
The abundance of ITS1 and ITS2 barcodes in analyzed samples expressed as the number of samples where the given taxon was determined.

**Figure 2 plants-12-02351-f002:**
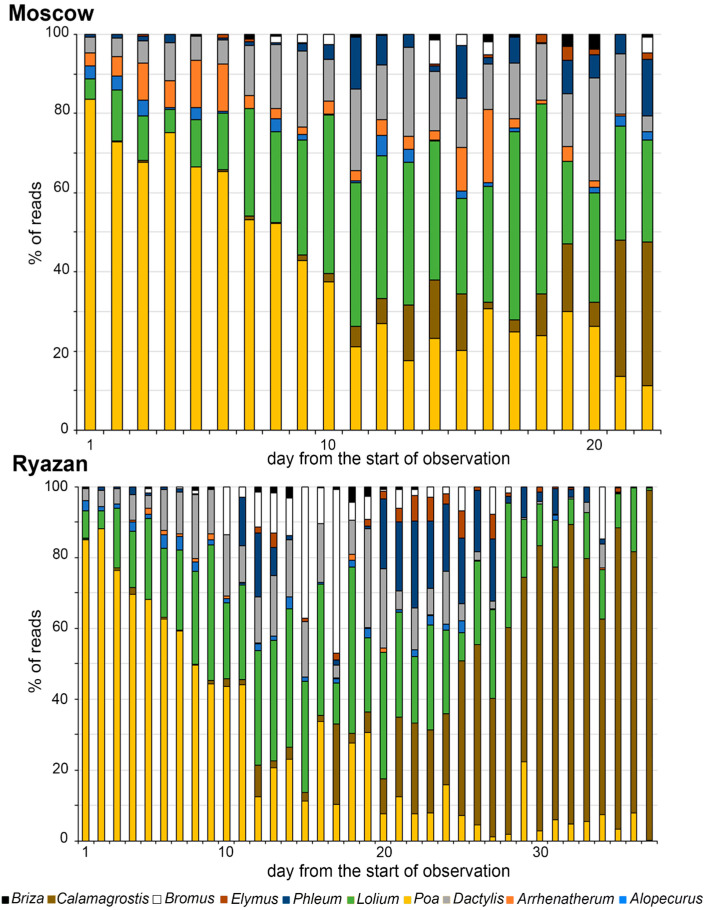
Average results for the metabarcoding analysis (ITS1 + ITS2) of airborne pollen samples from Moscow and Ryazan, 2020–2022.

**Figure 3 plants-12-02351-f003:**
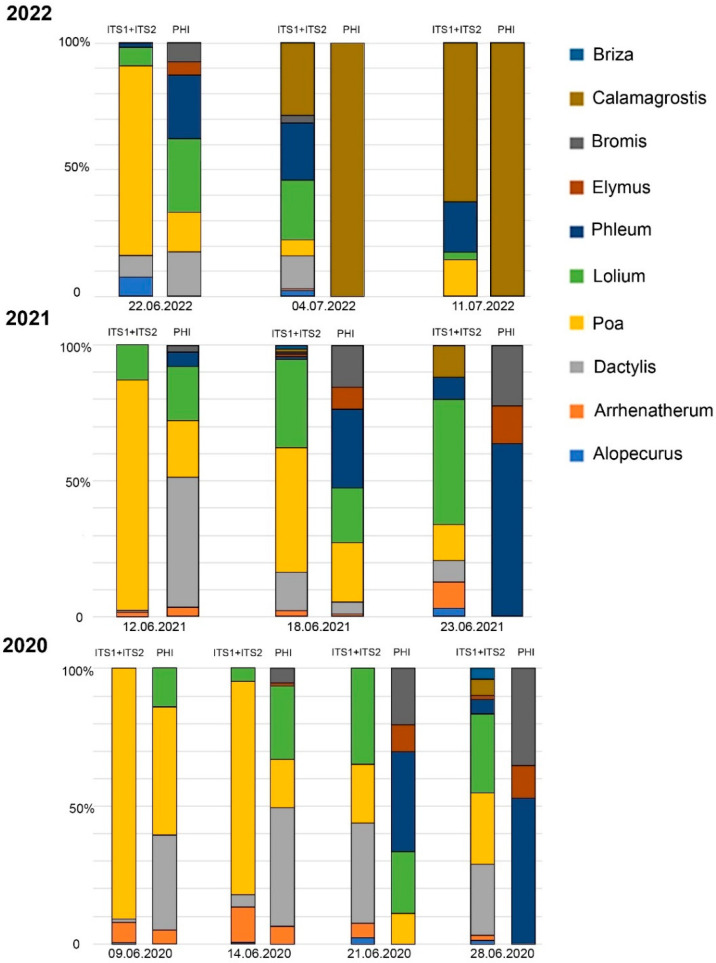
Comparison of metabarcoding results and phenological observations, Moscow, 2020–2022. PHI—phenological index.

**Figure 4 plants-12-02351-f004:**
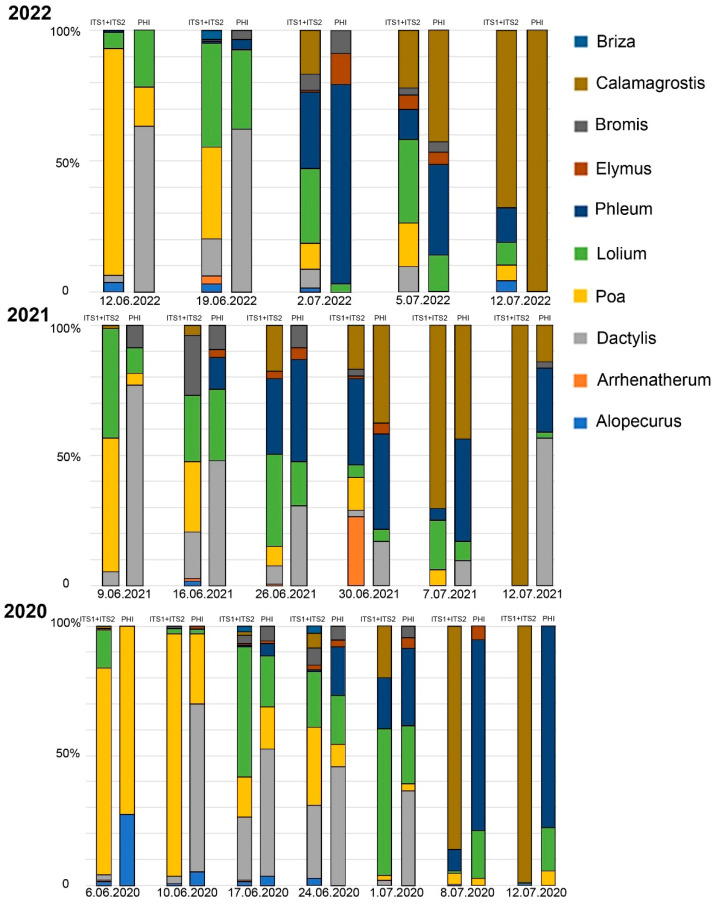
Comparison of metabarcoding results and phenological observations, Ryazan, 2020–2022. PHI—phenological index.

**Table 1 plants-12-02351-t001:** The efficiency of different methods of DNA isolation.

Method	C, ng/µL	OD 260/280	OD 260/230
1	0.6 ± 0.12	1.78 ± 0.28	0.24 ± 0.09
2	0.75 ± 0.52	1.62 ± 0.16	0.56 ± 0.27
3	0.71 ± 0.13	1.75 ± 0.19	0.57 ± 0.48
4a	0.42 ± 0.02	1.16 ± 0.67	0.68 ± 0.55
4b	0.94 ± 0.69	1.77 ± 0.2	0.78 ± 0.52
4c	1.13 ± 0.23	1.89 ± 0.13	2.09 ± 0.75

## Data Availability

All the sequenced data are deposited in the public GenBank database.

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
