# Peer review of "Aerobiological Monitoring and Metabarcoding of Grass Pollen"

_plants, 2023, doi:10.3390/plants12122351_

Round 1
Reviewer 1 Report
This is a really interesting and well-conducted research comparing morphological and molecular data from aerobiological monitoring and combining aerobiology with phenological observations. The approach demonstrates that analyses are complementary and can be integrated to obtain the best resolution and information possible from the data. The application of combined methods is greatly appreciable.
The Authors are expert in the field and wrote previous work on pollen productivity related molecular aspects of Poaceae.
The paper is well-organised and presents new data. Below, some observations to be addressed to improve the quality of the paper.
In the Introduction, the Authors must point out that there is no 'direct' correspondence between vegetation cover (as shown by phytosociological surveys) and airborne pollen, even when only anemophilous species are considered as the major contributors to airborne pollen uptake. Moreover, as I understood well, they must explain that the molecular approach is fundamental to improve the qualitative analysis within the quali-quanti-daily data given by aerobiological morphological methods.
The paper by K.L. Bell, K.S. Burgess, J.C. Botsch, E.K. Dobbs, T.D. Read, B.J. Brosi - Quantitative and qualitative assessment of pollen DNA metabarcoding using constructed species mixtures Mol. Ecol., 28 (2019), pp. 431-455, 10.1111/mec.14840
should be also considered in the Discussion.
In the methods, the way ‘pollen’ was taken for molecular analysis should be described better.
Keywords: add ‘phenological releeves’ or something referring to that task; add reference to the geographical location of the sample collection
Abstract
Grass pollen ‘can be identified only at the family level ‘ = actually some large pollen can be discriminated as cereals
Introduction
The first sentence is quite similar to the opening sentence of the paper the Authors published on Plants recently: should you consider to change it a bit more?
“Pollen is one of the main causes of pollinosis, …” = grass pollen? Consider that this is the first sentence after two sentences dealing with Poaceae, and the subsequent dealing with Poaceae. Here you have to cite the Grass pollen as a main cause of pollinosis
“However, phenological observations in aerobiology are scarce, as they require deep botanical knowledge and are very labor intensive. “ = I suggest to delete “and are very labor intensive”, because it is not an intensive but only a ‘different’ labour. A field botanist works in floristic surveys, and this is not a difficult task for a specialist.
“Longhi et al. [43]showed the possibility of replacing routine analysis with light microscopy methods with quantitative PCR analysis ….” = please, consider to explain that airborne pollen monitoring is a quantitative-qualitative and temporal analysis; the promising results – here tested on 3 plant species – can help on the qualitative aspects of pollen monitoring.
Results
Fig. 2 should be improved and the image can be reduced
Fig. 3 is not so easy to read; the legend must report how the percentage of Y axis is calculated, and what period refers to the X axis (weekly data?).
Methods
“Brown-Blanque scale” = be aware, the name is French and must be written as Braun-Blanquet
“To assess the optimal incubation duration, we used samples with artificially applied pollen in the same (approximately) amount as observed on trap samples. “ = do you mean micrograms? how the amount was estimated? What species? Dry or wet status?
Author Response
We thank the anonymous reviewer for a careful reading of our work and for valuable comments.
Reviewer 1
This is a really interesting and well-conducted research comparing morphological and molecular data from aerobiological monitoring and combining aerobiology with phenological observations. The approach demonstrates that analyses are complementary and can be integrated to obtain the best resolution and information possible from the data. The application of combined methods is greatly appreciable.
The Authors are expert in the field and wrote previous work on pollen productivity related molecular aspects of Poaceae.
The paper is well-organised and presents new data. Below, some observations to be addressed to improve the quality of the paper.
In the Introduction, the Authors must point out that there is no 'direct' correspondence between vegetation cover (as shown by phytosociological surveys) and airborne pollen, even when only anemophilous species are considered as the major contributors to airborne pollen uptake. Moreover, as I understood well, they must explain that the molecular approach is fundamental to improve the qualitative analysis within the quali-quanti-daily data given by aerobiological morphological methods.
We are thankful for the comments. We are not aiming to discuss the correspondence between the pollen spectrum and vegetation. As the reviewer has indicated this is a special and very complicated problem to which a vast literature is dedicated. Many different approaches are used to understand these relations and molecular approach is only one of them. It faces many difficulties in solving specific problems and today cannot be recognized as fundamental.
The paper by K.L. Bell, K.S. Burgess, J.C. Botsch, E.K. Dobbs, T.D. Read, B.J. Brosi - Quantitative and qualitative assessment of pollen DNA metabarcoding using constructed species mixtures Mol. Ecol., 28 (2019), pp. 431-455, 10.1111/mec.14840
should be also considered in the Discussion.
The work was added
In the methods, the way ‘pollen’ was taken for molecular analysis should be described better.
Pollen was collected with a volumetric trap. The samples (pieces of Melinex tape) were washed in a buffer. Information about the procedure of optimization of incubation time was added in Material and methods
Keywords: add ‘phenological releeves’ or something referring to that task; add reference to the geographical location of the sample collection
Keywords were added.
Abstract
Grass pollen ‘can be identified only at the family level ‘ = actually some large pollen can be discriminated as cereals
Pollen more than 40 mkm in diameter can be determined as “cereal group” in pollen and spore analysis, but real cereal species (Triticum, Secale, Zea etc) are extremely rare (if any) in the airborne pollen spectrum of our region. In aerobiological works, grass pollen is usually determined to the family level. We added clarification in the text
Introduction
The first sentence is quite similar to the opening sentence of the paper the Authors published on Plants recently: should you consider to change it a bit more?
“Pollen is one of the main causes of pollinosis, …” = grass pollen? Consider that this is the first sentence after two sentences dealing with Poaceae, and the subsequent dealing with Poaceae. Here you have to cite the Grass pollen as a main cause of pollinosis
The sentence was changed.
“However, phenological observations in aerobiology are scarce, as they require deep botanical knowledge and are very labor intensive. “ = I suggest to delete “and are very labor intensive”, because it is not an intensive but only a ‘different’ labour. A field botanist works in floristic surveys, and this is not a difficult task for a specialist.
Thank you for the comments, the sentence was changed.
“Longhi et al. [43]showed the possibility of replacing routine analysis with light microscopy methods with quantitative PCR analysis ….” = please, consider to explain that airborne pollen monitoring is a quantitative-qualitative and temporal analysis; the promising results – here tested on 3 plant species – can help on the qualitative aspects of pollen monitoring.
We have modified the sentence.
Results
Fig. 2 should be improved and the image can be reduced
The image was improved and reduced.
Fig. 3 is not so easy to read; the legend must report how the percentage of Y axis is calculated, and what period refers to the X axis (weekly data?).
We added the explanation of axis Y to the legend. Axis X does not indicate the period, it shows the dates of sampling. We did not use all samples for molecular analysis. As it was explained in Material and methods, the metagenomic analysis included only samples in which the total concentration of grass pollen was at least 50 pollen grains/m3 according to aerobiological observations. We changed the axis and place the list of samples in Supplementary Materials
axis Y - number of reads, in percents; axis X - date of sampling
Methods
“Brown-Blanque scale” = be aware, the name is French and must be written as Braun-Blanquet
Corrected
“To assess the optimal incubation duration, we used samples with artificially applied pollen in the same (approximately) amount as observed on trap samples. “ = do you mean micrograms? how the amount was estimated? What species? Dry or wet status?
A mix of dry pollen of 8 grass species was applied on a sticky surface of Melinex tape. The amount of pollen was controlled under a light microscope and calculated according to the method applied for aerobiological slides. Then the tape was divided into two parts - the first one was used as control and the other one was placed into the buffer. The number of pollen after washing was also controlled under a light microscope. The explanation was added to Material and methods.

Reviewer 2 Report
Dear Authors,
See my revision of your manuscript as a pdf file.
Reviewer

Author Response
We thank the anonymous reviewer for a careful reading of our work and for valuable comments.
Reviewer 2
How do you explain the monthly and interannual changes in barcode abundance of different Poaceae taxa?
The abundance of different barcodes in the air samples depends on many reasons as
1 the abundance of a particular plant in the region of sampling, its phenophase and pollen production in a particular season.
2 meteorological conditions during sampling period, the processes of secondary lifting and long-distance transport of pollen
3 amplification efficiency - GC-content and length variability can affect PCR efficiency
Are these changes statistically significant?
We did not carry out statistical analysis, since there is not enough data for each individual species in each season.
Can any system be observed in the taxa-dependent barcode abundance?
The abundance of a particular barcode reflects the abundance of the corresponding plant in the region and its phenophase. The comparison of metabarcoding results and phenological observation supports this idea.
Are there regional differences within and between Moscow and Ryazan regions?
We could not investigate the difference within regions as we have only one point of sampling in each region. It was shown earlier (Severova, Volkova 2018) that one pollen trap installed at the height of 10–12 m above ground level on the Central Russian Plain can reflect pollen diversity within a radius of 50 km. We believe that in each region we have studied the regional spectrum. The difference between the qualitative composition of the airborne pollen spectrum of Moscow and Ryasan is also minimal if any (Posevina, Severova, 2017, in Russ.). It may be the difference in the abundance of particular species but it cannot be revealed by the applied method.
